# Effect of Blade Size on the First-Pass Success Rate of Endotracheal Intubation Using the C-MAC Video Laryngoscope

**DOI:** 10.3390/jcm12227055

**Published:** 2023-11-13

**Authors:** Jeongyong Park, Goeun Park, Da Seul Kim, Minha Kim, Sejin Heo, Daun Jeong, Hansol Chang, Se Uk Lee, Goosang Choi, Gun Tak Lee, Tae Gun Shin, Jong Eun Park, Sung Yeon Hwang

**Affiliations:** 1Department of Emergency Medicine, Samsung Medical Center, Sungkyunkwan University School of Medicine, Seoul 06355, Republic of Korea; 2Biomedical Statistics Center, Research Institute for Future Medicine, Samsung Medical Center, Seoul 06351, Republic of Korea; 3Department of Digital Health, Samsung Advanced Institute for Health Science & Technology (SAIHST), Sungkyunkwan University, Seoul 06355, Republic of Korea; 4Department of Emergency Medicine, College of Medicine, Kangwon National University, Chuncheon 20341, Republic of Korea

**Keywords:** intubation, intratracheal, laryngoscope size, laryngoscopy, emergency

## Abstract

We sought to determine whether blade size influences the first-pass success (FPS) rate when performing endotracheal intubation (ETI) with a C-MAC video laryngoscope (VL) in emergency department (ED) patients. This single-center, retrospective, observational study was conducted between August 2016 and July 2022. A total of 1467 patients was divided into two categories based on the blade size used during the first ETI attempt: blade-3 (n = 365) and blade-4 groups (n = 1102). The primary outcome was the FPS rate. The secondary outcomes included the glottic view, multiple attempt rate, and ETI-related complications. We used propensity score matching to reduce the potential confounders between the two groups. Among these, 363 pairs of matched propensity scores were generated. The FPS rate did not differ between the blade-3 (84.8%) and blade-4 groups (87.3%) in the matched cohort (*p* = 0.335). The multiple attempt rate did not differ significantly between groups (*p* = 0.289) and was 3.9% and 2.5% in the blade-3 and blade-4 groups, respectively. The difficult glottic view (11.3 vs. 6.9%, *p* = 0.039) and complication rates (15.4% vs. 10.5%, *p* = 0.047) were significantly higher in the blade-3 group than in the blade-4 group. The FPS rates of ETI with the blade-3 and blade-4 groups in adult patients in the ED did not differ significantly.

## 1. Introduction

Endotracheal intubation (ETI) is a life-saving procedure that is frequently performed in critically ill or injured patients for airway protection, oxygenation, and ventilation [1]. Traditionally, direct laryngoscope (DL) has been the standard device for ETI [2]. While DL continues to be widely utilized in clinical practice, there has been a notable surge in the popularity of video laryngoscopes (VL) [3], primarily because they provide real-time visualization of the glottis, enhancing the glottic view and first-pass success (FPS) rate [4,5]. During the COVID-19 pandemic, the use of VL further expanded because most guidelines recommended using VL from the first attempt due to its increased distance from the patient’s mouth, making the procedure safer than using DL [6,7].

ETIs are associated with a high risk of adverse events including aspiration, cardiovascular collapse, severe hypoxemia, and cardiac arrest [1,8]. Several variables were associated with major peri-intubation adverse events. This involves not only patient-related factors, such as age, medical history, pre-intubation hemodynamics, and indications for ETI, but also a multitude of procedure-related factors, including the operator’s level of expertise, medications, and devices utilized during ETI [9,10]. In addition, several studies have shown the importance of achieving FPS, as the risk of adverse events significantly increases with the increasing number of ETI attempts [11,12]. Consequently, considerable efforts have been made to identify techniques and measures associated with improving the FPS rate.

Currently, there are a lack of data regarding the selection of blade size, forcing clinicians to rely on a combination of device availability, patient-specific characteristics, and the preferences of individual physicians. Recently, some studies have suggested that increasing the FPS rates through the selection of an appropriate blade size could be a straightforward approach that does not require additional interventions [13,14,15]. However, these studies solely assessed the Macintosh-type DL, and no study has examined the link between the success rate of ETI and the VL blade size. A distinction exists in the way each piece of equipment is operated between both approaches. DL requires the displacement of the upper airway tissues to establish a straight line of sight extending from the oral cavity to the laryngeal inlet, while VL obviates the need to align the axes by utilizing an image sensor located in the distal portion of the blade to provide a panoramic view of the glottis. Therefore, it may be unsuitable to extrapolate data from the DL to the VL, representing a knowledge gap regarding the relationship between the FPS rate and blade size of the VL. Furthermore, it is of clinical significance to establish this association, considering the increasing prevalence of VL as a standard device for ETI.

The objective of this study was to assess the impact of blade size on the FPS rate during ETI using VL in adult patients treated in the emergency department (ED).

## 2. Materials and Methods

### 2.1. Study Design and Setting

This single-center, retrospective, observational study was conducted in the academic ED of a university-affiliated tertiary referral hospital between August 2016 and July 2022. The ED treats more than 70,000 patients each year. Approximately 400 ETIs are performed annually in adult patients in the ED, most of which are performed by emergency medicine (EM) physicians. Each ETI procedure performed in the ED was supervised by an EM faculty member. Patients resistant to airway manipulation with laryngoscopy and placement of an endotracheal tube were treated with rapid sequence intubation as the standard treatment. A semi-rigid stylet was routinely used to facilitate the insertion of the endotracheal tube into the trachea. The Institutional Review Board of Samsung Medical Center approved the study (number 2023-04-003) and waived the requirement for informed consent because the study was retrospective and used anonymous data. All the methods were carried out in accordance with relevant institutional guidelines and regulations.

### 2.2. Selection of Participants

Patients who met the following criteria were included in the analysis: (1) adult patients (18 years or older) who underwent ETI in the ED and (2) patients for whom the blade-3 (C-MAC VL MAC #3) or blade-4 (MAC #4) (Karl Storz Endoscope, Tuttlingen, Germany) groups were used in the first attempt. The exclusion criteria were as follows: (1) ETI performed with a device other than C-MAC VL MAC #3 or MAC #4, such as a direct laryngoscope or D-BLADE (Karl Storz Endoskope, Tuttlingen, Germany); (2) lack of data regarding the size of the blade used; and (3) endotracheal tube exchange with the tube exchanger. The patients were divided into two groups based on the blade size used during the first ETI attempt. The blade-3 group consisted of those intubated with C-MAC VL MAC #3, and the blade-4 group consisted of those intubated with C-MAC VL MAC #4.

### 2.3. Data Collection

Data were obtained from our institution’s electronic medical records and the institutional airway registry for quality improvement in airway management [16,17]. To limit recall bias, all ETI processes were recorded in real time by a staff member who did not participate in the procedure. The records were verified immediately after ETI by the intubator and the faculty member in charge to ensure that the data were correct. The following data were collected for analysis: patient age, sex, height, weight, vital signs, peripheral oxygen saturation at the time of intubation decision, indications for ETI, ETI methods including crash approach, rapid sequence intubation, sedative-only approach, presence of difficult laryngoscopy and ETI characteristics, intubating device, blade size, drugs for ETI, specialty and training level of the intubator, number of ETI attempts, success of ETI, glottic view indicated by Cormack–Lehane (C-L) grade, and complications related to ETI.

### 2.4. Definition

The crash approach was defined as attempting ETI without the use of medication in a collapsed or unresponsive patient who was not anticipated to be resistant to laryngoscopy and required immediate airway security. Rapid sequence intubation involves preoxygenation, followed by the administration of a potent induction agent virtually simultaneously with a rapidly acting neuromuscular blocking agent to induce sedation and muscle paralysis, thereby facilitating ETI. Based on the training years, the intubators were divided into three categories: junior residents (first- or second-year residents), senior residents (third- or fourth-year residents), and EM specialists. The intubator and supervisory staff determined the presence of difficult airway characteristics for laryngoscopy and ETI based on the patient’s anatomical features, including obesity, short neck, distorted airway anatomy, facial trauma or anomaly, limited mouth opening (3 cm), and cervical immobility. The intubator determined the glottic view using the C-L classification system. A C-L grade of three or four was defined as a difficult glottic view. An ETI attempt was defined as the insertion of a laryngoscope blade into the airway, irrespective of the insertion of an endotracheal tube. If another intubator took over the laryngoscope handle while the laryngoscope blade was in the patient’s mouth, this was counted as one attempt. FPS and multiple attempts were defined as passing the ETI on the initial attempt and attempting ETI three or more times, respectively.

### 2.5. Primary and Secondary Outcomes

The primary outcome measure was the FPS rate. The secondary outcomes included a difficult glottic view on the first attempt, the multiple attempt rate, and ETI-related complications.

### 2.6. Primary Data Analysis

Based on the results of the normality test, continuous variables are presented as the means and standard deviations or medians and interquartile ranges. An independent two-sample *t*-test was used to compare the means, and the Wilcoxon rank-sum test was used to compare the medians. The number of cases and percentages are presented as representative values for categorical variables, and the variables were compared using the chi-square test or Fisher’s exact test, as applicable.

We used propensity score matching (PSM) between the blade-3 and blade-4 groups to reduce the potential confounders. The following variables were used as matching variables: age, sex, height, weight, body mass index (BMI), indications for ETI, ETI methods (crash approach, rapid sequence intubation, and sedation-only), presence of difficult laryngoscopy and ETI characteristics, and intubator level at the first attempt. A logistic regression model was used to estimate the propensity score for each patient, representing the predicted probability of the blade size-4 group. We performed 1:1 matching using the nearest-neighbor greedy matching method with a caliper of 0.2. A standardized mean difference of <0.2 between the blade-3 and blade-4 groups was used to evaluate matching adequacy.

Univariable and multivariable logistic regression analyses were performed to evaluate the relationship between the blade size and FPS rate. The following variables were included as confounding factors: age, height, weight, BMI grade, sex, method of ETI, presence of anticipated difficult airway characteristics, and level of intubator.

Subsequently, the cohort was divided into five strata according to the quintiles of the estimated propensity scores. Stratified logistic regression analysis was performed to produce an overall OR considering the strata. Logistic regression analyses were performed separately within each stratum to examine the relationship between the blade size and FPS rate.

In addition, logistic regression models were built using propensity scores to adjust for differences among groups in three alternative ways: (1) regression adjustment by including the propensity score as a covariate in the regression model; (2) use of the propensity score to create stabilized weights, defined as the inverse probability of treatment weighting; and (3) propensity score matching (conditional logistic regression was performed to assess the association between the blade size and FPS using 1:1 PSM data).

Subgroup analyses were also conducted. Subgroups were established based on the following criteria: age (<65 years old vs. ≥65 years old), sex (female vs. male), height (<170 cm vs. ≥170 cm; <175 cm vs. ≥175 cm), presence of difficult airway characteristics for laryngoscopy and ETI (yes vs. no), and level of intubator (junior vs. non-junior). For subgroup analysis, the intubators were divided into two categories: junior (first- or second-year residents) and non-junior (third- or fourth-year residents and EM specialists). During the subgroup analysis, the following variables were adjusted within each subgroup: age, height, weight, BMI grade, sex, methods for ETI, presence of anticipated difficult airway characteristics, and level of intubator. The variables corresponding to each subgroup were eliminated from the adjustments.

Statistical significance was set at a *p*-value of <0.05. SAS version 9.4 (SAS Institute, Cary, NC, USA) and PASS 2022 (version 22.0.2, NCSS, LLC., Kaysville, UT, USA) were used for the statistical analysis.

## 3. Results

### 3.1. Baseline Characteristics

During the study period, 2685 adult patients underwent ETI. The primary analysis included 1467 patients after excluding 1218 patients who met the exclusion criteria. The patients were divided into the blade-3 (n = 365) and blade-4 (n = 1102) groups. Among these patients, 363 propensity score-matched pairs were generated (1-to-1 matching: blade-3 group (n = 363) vs. blade-4 group (n = 363)) (Figure 1).

Table 1 shows the baseline characteristics of the matched and unmatched patients and ETI-related variables according to the propensity scores (also see Appendix A).

In the unmatched cohort, the male sex was more frequent in the blade-4 group (67.8%) compared to the blade-3 group (48.8%) (*p* < 0.001). Patient BMI was significantly higher in the blade-4 group (23.3 ± 3.9) than in the blade-3 group (22.3 ± 3.6) (*p* < 0.001). The most common indication for ETI in all groups was cardiac arrest, followed by respiratory distress, altered mental status, and shock, with no significant differences between the groups. The presence of difficult airway characteristics was not significantly different between the groups (blade-3 group vs. blade-4 group: n = 91 (24.9%) vs. n = 303 (27.5%); *p* = 0.338). The first attempt was made mostly by EM physicians (91.5% in the blade-3 group vs. 95.3% in the blade-4 group, *p* = 0.007). As shown in Table 1, the variables with large, standardized differences before matching were reduced after matching, resulting in balanced baseline characteristics.

### 3.2. Primary and Secondary Outcomes

The primary and secondary outcomes are presented in Table 2. In the unmatched cohort, the FPS rate was not significantly different between the two groups (84.9% in the blade-3 group vs. 86.8% in the blade-4 group, *p* = 0.357). The multiple attempt rate was not significantly different between the two groups (3.8% in the blade-3 group and 3.5% in the blade-4 group, *p* = 0.792). There was a significantly higher incidence of a difficult glottic view in the blade-3 group (11.2%) compared to the blade-4 group (7.0%) (*p* = 0.010). There was no significant difference in the overall complication rate between the groups (15.3% in the blade-3 group vs. 12.4% in the blade-4 group, *p* = 0.154).

In the matched cohort, the FPS rate was not different between the blade-3 group (84.8%) and the blade-4 group (87.3%) (*p* = 0.335). The multiple attempt rate was not significantly different between the two groups (3.9% in the blade-3 group and 2.5% in the blade-4 group, *p* = 0.289). The incidence of a difficult glottic view (11.3% vs. 6.9%, *p* = 0.039) and the rate of complications (15.4% vs. 10.5%, *p* = 0.047) were significantly higher in the blade-3 group than in the blade-4 group.

### 3.3. Effect of Blade Size on the Odds Ratios for First-Pass Success Rate

Table 3 shows the effect of blade size on the odds ratios for the FPS rate. The FPS between the two groups did not differ significantly in the multivariable logistic regression model (adjusted OR, 1.236; 95% CI, 0.864–1.767; *p* = 0.246). The superiority of a particular blade size was not supported by any of the propensity score-adjusted models (all *p* > 0.05).

In the subgroup analyses based on age (<65 years old vs. ≥65 years old), male sex, height (<170 cm vs. ≥170 cm; <175 cm vs. ≥175 cm), presence of difficult airway characteristics for laryngoscopy and ETI (yes vs. no), and level of intubator (junior residents vs. non-junior residents) (all *p* > 0.05), the blade size was not significantly associated with the FPS rate (all *p* > 0.05). However, in the case of the female group, the odds ratio for FPS in the blade-4 group compared with the blade-3 group was 1.755 (95% CI, 1.040–2.962; *p* = 0.035), indicating that using blade-4 instead of blade-3 may result in a higher FPS rate. 

### 3.4. Effect Size

There were 1102 patients in the blade-4 group and 365 in the blade-3 group. The minimum detectable effect size, under the assumption of an 85% FPS rate for the blade-3 group, was 5.3%, with a significance level of 0.05 and a power of 0.8. The FPS rates for the blade-3 and blade-4 groups in our study were 84.9% and 86.8%, respectively, prior to PSM, and 84.8% and 87.3%, respectively, after PSM. Therefore, these differences were not statistically significant.

## 4. Discussion

In this study, we compared the performance of C-MAC VL MAC #3 and MAC #4 in facilitating ETI in adult patients in an ED. Based on the available data, this is the first study on the relationship between the blade size of a VL and the FPS rate. We found no significant difference in the FPS rates between the blade-3 and blade-4 groups before and after PSM. There was no significant association between the blade size and FPS rate in several propensity score-adjusted models and the multivariable logistic regression model. In addition, there were no significant differences in the multiple attempt rate between the two groups. The blade-4 group had a better glottic view rate and a lower rate of ETI-related complications after PSM than the blade-3 group. However, the improved glottic view rate did not translate into an increase in the FPS rate. In addition, the higher overall complication rate was primarily due to post-intubation hypotension which is unlikely to be influenced by blade size. Therefore, this study may not support the selection of a specific blade size when C-MAC VL is used for ETI in the ED.

Several studies on the effect of the Macintosh-type blade size on the success rate of ETI when using DL found improved outcomes with a blade size of three compared with a blade size of four [14,15]. However, these studies focused on DL, not VL. Godget et al. compared Macintosh DL blade sizes of three and four using DL in 2139 patients in French intensive care units [14]. In this study, a blade size of three was associated with increased FPS (79.5% versus 73.3%; relative risk, 1.41; 95% confidence interval [CI], 1.23–1.77; *p* = 0.0025, respectively) without any difference in the complication rate. In a study by Landefeld et al., Macintosh DL using blade sizes of three and four were compared for the first ETI attempt in 584 patients from the ED and intensive care unit [15]. Patients intubated with a blade size of four had a worse C-L grade in the glottic view (adjusted odds ratio (aOR), 1.458; 95% CI, 1.064–2.003; *p* = 0.02) and lower FPS (71.1% vs. 81.2%; aOR, 0.566; 95% CI, 0.372–0.850; *p* = 0.01) compared to those intubated with a blade size of three. These findings support the use of a Macintosh blade size of three for ETI in critically ill patients undergoing DL. Potential explanations for these findings have been proposed [13,14,15,18]. The longer length and wider width of the blade with a blade size of four compared to three may influence the extent of exposure and visualization of the laryngeal structures. A larger blade may require more space in the oral cavity, potentially limiting the room available for the manipulation and maneuvering of the blade and endotracheal tube. This can make it more challenging to achieve the optimal alignment of the oral, pharyngeal, and laryngeal axes necessary for successful intubation. In this regard, Landefeld et al. reported that a blade size of four provided a poorer glottic view than a blade size of three [15]. In addition, Kim et al. found that a blade size of three performed better for novices than a blade size of four because a higher esophageal visualization rate with a blade size of four led to esophageal intubation [13].

In contrast to previous findings, the blade size had no significant effect on the FPS rate or multiple attempts in this study, and the glottic view was better with a blade size of four than with a blade size of three. In VL, the airway is displayed on a monitor, which provides a magnified and clearer view of the laryngeal structures and enables multiple staff members to share it in real time. This facilitates the supervision and guidance of experienced physicians throughout the ETI procedure. In addition, although an improved glottic view did not translate into an increase in the FPS in our study, the blade-4 group showed a better glottic view than the blade-3 group. These factors may have compensated for the limited maneuverability of the larger blade as well as its size-related vision impairment. The relatively high FPS rate in our study (84.9% in the blade-3 group vs. 86.8% in the blade-4 group) compared with that in previous studies (study by Landefeld et al., 81.2% in the Macintosh size 3 vs. 71.1% in the Macintosh size 4; study by Godet et al., 84.1% in the Macintosh size 3 vs. 72.1% in the Macintosh size 4), which compared the difference in DL by blade size, may also have contributed to the inability to demonstrate a difference in FPS by blade size [14,15].

From a clinical perspective, it would be meaningful to identify the factors that could predict the appropriate blade size for each patient. No studies have identified the factors that can reliably predict the optimal blade size for a given patient. Except for females, subgroup analyses revealed no association between the blade size and FPS rate. A higher FPS rate was associated with the use of a blade size of four, as opposed to three, among females. Inamoto et al. reported that the upper airway volume of females was smaller than that of males, and this difference persisted even after adjusting for height; that is, females had a smaller upper airway volume than males even when they had the same height [19]. Paradoxically, the use of a larger blade was associated with a higher FPS rate in females. The rationale for this finding remains unclear, and providing an explanation for its origin is beyond the scope of this study. However, the findings of this study suggest that the improvement in the FPS rate may be affected by the selection of a blade of a particular size depending on specific factors. Clarification of the factors that can be used as predictors of the optimal blade size requires further research.

This study has several limitations. First, because our study was a retrospective analysis, the baseline characteristics of the patients and ETI-related factors could not be controlled. In particular, the selection of the blade size was dependent on the device availability or the intubator’s preference. Moreover, a blade size of four was utilized more often than a blade size of three at our institution, which may have facilitated the intubators’ familiarity with a blade size of four. Although we used PSM analysis to limit the likelihood of bias, undetected confounders might have had an impact on the results. Second, this was a single-center study conducted in a university-affiliated ED; therefore, the results may not apply to other contexts. Third, because only the blade size of C-MAC VL was evaluated in this study, the results may not be applicable to other equipment. Due to the different VL configurations (e.g., channeled vs. hyperangulated vs. standard geometry blade), caution needs to be taken when extrapolating our findings to other devices. Finally, the study population was relatively short in height and comprised a small proportion of obese patients [15]. Consequently, the applicability of our findings to other settings in which a substantial proportion of patients are taller or obese may be limited. To address these concerns, this study should be replicated in areas with a significant proportion of tall and obese patients.

## 5. Conclusions

In conclusion, the FPS rates for ETI with C-MAC VL MAC #3 and C-MAC VL #4 in adult patients in the ED did not differ significantly. In addition, there were no substantial differences between the two groups in terms of multiple attempts. Although C-MAC VL #4 provided a superior glottic view, this did not translate into an increase in the FPS rate. To determine the effect of blade size on the success rate of ETI in VL, a large-scale study using multiple devices is required.

## Figures and Tables

**Figure 1 jcm-12-07055-f001:**
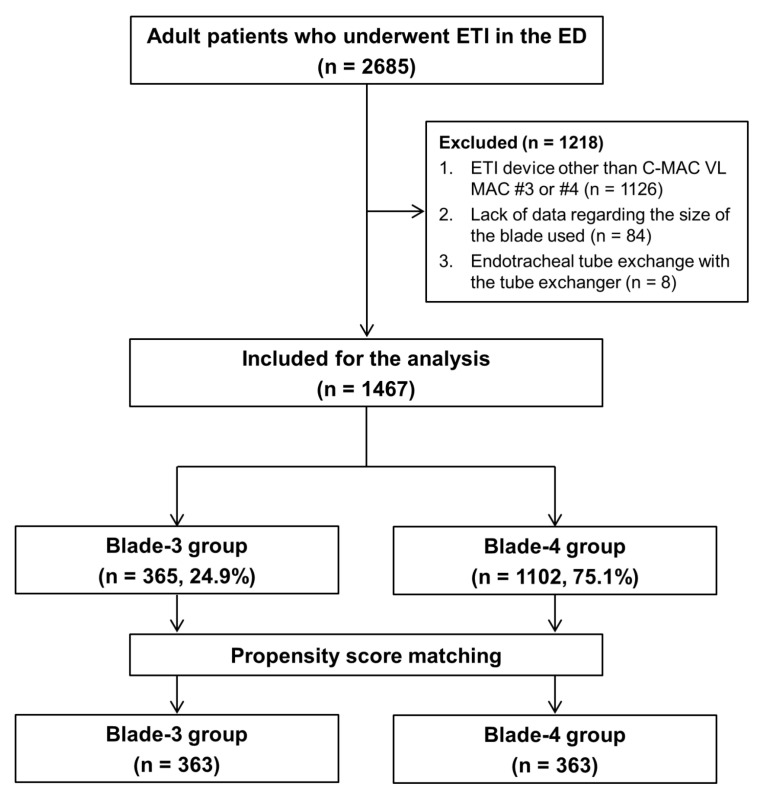
Study flow chart. ETI, endotracheal intubation; ED, emergency department; VL, video laryngoscope.

**Table 1 jcm-12-07055-t001:** Baseline characteristics of the study population and endotracheal intubation-related variables.

	Unmatched Cohort				Propensity Score-Matched Cohort *			
	Blade-3 Group (n = 365)	Blade-4 Group (n = 1102)	*p*-Value	Standardized Difference	Blade-3 Group (n = 363)	Blade-4 Group (n = 363)	*p*-Value	Standardized Difference
Patient sex (male) *	178 (48.8)	747 (67.8)	<0.001	−0.393	178 (49.0)	169 (46.6)	0.504	0.050
Patient age (years) *	66.8 ± 16.0	65.2 ± 15.8	0.089	0.103	66.7 ± 16.0	66.5 ± 16.8	0.842	0.015
Patient height *	162.5 ± 8.7	165.8 ± 9.1	<0.001	−0.376	162.6 ± 8.5	162.3 ± 9.3	0.688	0.030
Patient weight *	59.3 ± 12.6	64.4 ± 13.9	<0.001	−0.386	59.4 ± 12.5	59.7 ± 12.1	0.814	−0.018
Patient BMI, (kg/m^2^)	22.3 ± 3.6	23.3 ± 3.9	<0.001	−0.252	22.3 ± 3.6	22.5 ± 3.5	0.526	−0.047
Patient BMI grade *			0.002				0.792	
Under (below 18.5)	45 (12.3)	108 (9.8)		0.081	45 (12.4)	43 (11.8)		0.017
Normal (18.5–24.9)	252 (69.0)	679 (61.6)		0.157	250 (68.9)	241 (66.4)		0.053
Over (25.0–29.9)	58 (15.9)	261 (23.7)		−0.197	58 (16.0)	67 (18.5)		−0.066
Obese (30.0 and above)	10 (2.7)	54 (4.9)		−0.113	10 (2.8)	12 (3.3)		−0.032
Vital signs and SpO_2_ †								
Systolic blood pressure (mmHg)	128.1 ± 45.0	128.1 ± 48.6	0.987	−0.001	128.3 ± 45.1	124.8 ± 44.2	0.416	0.077
Diastolic blood pressure (mmHg)	69.8 ± 25.1	70.3 ± 29.3	0.810	−0.018	70.0 ± 25.2	68.4 ± 27.4	0.532	0.060
Heart rate (per minute)	110.6 ± 25.9	110.1 ± 29.8	0.786	0.020	110.5 ± 26.0	111.7 ± 29.5	0.663	−0.042
Respiratory rate (per minute)	26.8 ± 8.4	26.13 ± 8.81	0.313	0.079	26.8 ± 8.4	25.9 ± 8.5	0.245	0.111
SpO_2_ (%)	95 (89–100)	96 (89–100)	0.604	0.101	95 (89–100)	96 (88–100)	0.558	0.066
ETI indication *			0.340				0.848	
Cardiac arrest	131 (35.9)	376 (34.1)		0.037	131 (36.1)	134 (36.9)		−0.017
Altered mental status	63 (17.3)	224 (20.3)		−0.079	63 (17.4)	72 (19.8)		−0.064
Respiratory distress	122 (33.4)	373 (33.8)		−0.009	122 (33.6)	109 (30.0)		0.077
Shock	44 (12.1)	104 (9.4)		0.085	42 (11.6)	43 (11.8)		−0.009
Others	5 (1.4)	25 (2.3)		−0.067	5 (1.4)	5 (1.4)		0.000
Methods for ETI *			0.147				0.849	
Crash approach ‡	142 (38.9)	406 (36.8)		0.043	142 (39.1)	142 (39.1)		0.000
RSI	191 (52.3)	627 (56.9)		−0.092	191 (52.6)	195 (53.7)		−0.022
Sedative only	32 (8.8)	69 (6.3)		0.095	30 (8.3)	26 (7.2)		0.041
Anticipated difficult airway *	91 (24.9)	303 (27.5)	0.338	−0.058	91 (25.1)	98 (27.0)	0.554	−0.044
Manual assist when intubating	22 (6.0)	74 (6.7)	0.645	−0.028	22 (6.1)	23 (6.3)	0.878	−0.011
First attempt								
Specialty of the intubator			0.007				0.150	
EM	334 (91.5)	1050 (95.3)		−0.152	332 (91.5)	342 (94.2)		−0.107
Non-EM	31 (8.5)	52 (4.7)		0.152	31 (8.5)	21 (5.8)		0.107
Level of intubator *			0.647				0.380	
Junior resident §	190 (52.1)	601 (54.5)		−0.050	189 (52.1)	186 (51.2)		0.017
Senior resident	141 (38.6)	411 (37.3)		0.028	141 (38.8)	153 (42.1)		−0.067
EM specialist	34 (9.3)	90 (8.2)		0.041	33 (9.1)	24 (6.6)		0.092
Sedatives			0.233				0.355	
Ketamine	69 (31.9)	176 (26.6)		0.117	67 (31.3)	53 (25.2)		0.135
Etomidate	134 (62.0)	455 (68.8)		−0.143	134 (62.6)	143 (68.1)		−0.115
Midazolam	12 (5.6)	28 (4.2)		0.061	12 (5.6)	14 (6.7)		−0.044
Others	1 (0.5)	2 (0.3)		0.026	1 (0.5)	0 (0.0)		0.097
NMBAs			0.910				0.836	
Succinylcholine	88 (45.4)	277 (44.0)		0.028	88 (45.4)	90 (46.2)		−0.016
Rocuronium	101 (52.1)	334 (53.0)		−0.019	101 (52.1)	102 (52.3)		−0.005
Others ¶	5 (2.6)	19 (3.0)		−0.027	5 (2.6)	3 (1.5)		0.073
Surgical approach	0 (0.0)	2 (0.2)	>0.999	−0.060	0 (0.0)	0 (0.0)	-	0.000
Failed to secure airway	0 (0.0)	2 (0.2)	>0.999	−0.060	0 (0.0)	0 (0.0)	-	0.000

The data are presented as the mean ± standard deviation, median with interquartile ranges, or number (%). * Propensity scores are matched. † Measured at the time of decision to intubate. ‡ Crash approach: Used for unconscious, unresponsive patients who were expected not to be resistant to the laryngoscopy and needed immediate airway security. § Junior resident refers to first- and second-year residents; senior resident refers to third- and fourth-year residents. ¶ Vecuronium and cisatracurium were included. Abbreviations: BMI, body mass index; SpO_2_, peripheral oxygen saturation; ETI, endotracheal intubation; RSI, rapid sequence intubation; EM, emergency medicine; NMBA, neuromuscular blocking agent.

**Table 2 jcm-12-07055-t002:** Outcomes.

	Unmatched Cohort			Propensity Score-Matched Cohort *		
	Blade-3 Group (n = 365)	Blade-4 Group (n = 1102)	*p*-Value	Blade-3 Group (n = 363)	Blade-4 Group (n = 363)	*p*-Value
First-pass success rate	310 (84.9)	957 (86.8)	0.357	308 (84.8)	317 (87.3)	0.335
Multiple attempts rate *	14 (3.8)	39 (3.5)	0.792	14 (3.9)	9 (2.5)	0.289
Glottic view			0.010			0.039
C-L Grade I or II	324 (88.8)	1025 (93.0)		322 (88.7)	338 (93.1)	
C-L Grade III or IV	41 (11.2)	77 (7.0)		41 (11.3)	25 (6.9)	
Complications						
Any complications	56 (15.3)	137 (12.4)	0.154	56 (15.4)	38 (10.5)	0.047
Post-intubation hypotension ‡	29 (7.9)	50 (4.5)	0.012	29 (8.0)	14 (3.9)	0.018
Post-intubation hypoxemia ‡	9 (2.5)	27 (2.5)	0.987	9 (2.5)	12 (3.3)	0.507
Vomiting ‡	0 (0.0)	2 (0.2)	>0.999	0 (0.0)	1 (0.3)	>0.999
Esophageal intubation	10 (2.7)	18 (1.6)	0.181	10 (2.8)	7 (1.9)	0.462
Unrecognized EI †	0 (0.0)	0 (0.0)	-	0 (0.0)	0 (0.0)	-
Arrhythmia ‡	2 (0.5)	2 (0.2)	0.260	2 (0.6)	1 (0.3)	>0.999
Agitation ‡	8 (2.2)	15 (1.4)	0.268	8 (2.2)	2 (0.6)	0.056
Dental injury	1 (0.3)	11 (1.0)	0.314	1 (0.3)	3 (0.8)	0.624
Pneumothorax	0 (0.0)	2 (0.2)	>0.999	0 (0.0)	0 (0.0)	-
Injury of airway	0 (0.0)	1 (0.1)	>0.999	0 (0.0)	1 (0.3)	>0.999
Cardiac arrest ‡	8 (2.2)	21 (1.9)	0.734	8 (2.2)	5 (1.4)	0.401
Death ‡	5 (1.4)	6 (0.5)	0.154	5 (1.4)	0 (0.0)	0.062

The data are presented as numbers (%). * Multiple attempts were defined as three or more intubation attempts. † Unrecognized EI was defined as esophageal intubation found after the patient’s condition worsened. ‡ These variables were observed in patients without cardiac arrest. Abbreviations: C-L grade, Cormack–Lehane grade; EI, esophageal intubation.

**Table 3 jcm-12-07055-t003:** Effect of blade size on the odds ratios for first-pass success rate.

		Blade-3 Group (n = 365)	Blade-4 Group (n = 1102)	OR (95% CI)	*p*-Value
Models					
Univariable logistic regression model		365	1102	1.171 (0.837–1.638)	0.357
Multivariable logistic regression model *		365	1102	1.236 (0.864–1.767)	0.246
PS-adjusted model					
Stratification		365	1102	1.179 (0.836–1.662)	0.348
Within-PS quintile					
	1 †	115	178	0.8 (0.318–2.015)	0.636
	2	85	209	1.043 (0.409–2.661)	0.929
	3	74	219	0.811 (0.335–1.96)	0.642
	4	45	249	1.499 (0.764–2.941)	0.239
	5 ‡	46	247	1.574 (0.811–3.055)	0.180
Regression adjustment		365	1102	1.174 (0.832–1.656)	0.362
Weighting (Stabilized IPTW)		365	1102	1.118 (0.809–1.547)	0.499
Matching 1:1		363	363	1.243 (0.806–1.917)	0.324
Subgroup analysis *					
Age (years)	<65	142	479	1.728 (0.955–3.124)	0.071
	≥65	223	623	0.986 (0.621–1.565)	0.952
Sex	Female	187	355	1.755 (1.040–2.962)	0.035
	Male	178	747	0.863 (0.506–1.471)	0.588
Height (cm)	<170	276	640	1.154 (0.748–1.780)	0.517
	≥170	89	462	1.325 (0.664–2.642)	0.425
Height (cm)	<175	321	845	1.224 (0.826–1.815)	0.314
	≥175	44	257	1.022 (0.379–2.756)	0.967
Difficult airway §	No	274	799	1.333 (0.839–2.117)	0.223
	Yes	91	303	1.172 (0.648–2.122)	0.599
Level of intubator ¶	Junior	190	601	1.248 (0.795–1.961)	0.336
	Non-junior	175	501	1.111 (0.598–2.064)	0.738

* The following confounding variables were included: age, height, weight, body mass index (BMI) grade, sex, indication for endotracheal intubation, methods of endotracheal intubation, presence of anticipated difficult airway characteristics, and level of intubator. During the subgroup analysis, the following variables were adjusted within each subgroup: difficult airway characteristics and level of intubator. The variables corresponding to each subgroup were eliminated from the adjustments. † Lowest propensity score. ‡ Highest propensity score. § Presence of difficult airway characteristics on laryngoscopy and endotracheal intubation. ¶ Junior refers to first- and second-year residents; non-junior refers to third- and fourth-year residents and emergency medicine specialists. Abbreviations: OR, odds ratio; CI, confidence interval; PS, propensity score; IPTW, inverse probability of treatment weighting.

## Data Availability

The datasets used and/or analyzed during the current study are available from the corresponding author upon reasonable request.

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
