# Peer review of "Effect of Blade Size on the First-Pass Success Rate of Endotracheal Intubation Using the C-MAC Video Laryngoscope"

_jcm, 2023, doi:10.3390/jcm12227055_

Round 1

Reviewer 1 Report

Comments and Suggestions for Authors

Dear authors,

The manuscript presented a valuable data on the relationship between blade size and first pass success.

I think the overall study design was appropriate, however, I would like the authors to elaborate on the effect size that needed to present the relationship. Please show what is the sample size for this study. How many people should be included into the study to see the statistical significance.

In the introduction part, line 63-64, there must be more factors affecting complications' risks of fail intubation, not only the number of attempt. Moreover, the knowledge gap in this area was not clear, the DL has been introduced for a very long time why should this manuscript be published. Please elaborate more on why do we need to add this information into the scientific knowledge?

Line 74, which types of blades did this manuscript focus on? VL or DL, some places you mentioned were DL, some were VL. Please revise to make the manuscript more congruent.

Comments on the Quality of English Language

Average

Author Response

C) I think the overall study design was appropriate, however, I would like the authors to elaborate on the effect size that needed to present the relationship. Please show what is the sample size for this study. How many people should be included into the study to see the statistical significance.

 (A) Thank you for your comments. We have added this information to the Results section.

“3.4 Effect size

There were 1,102 patients in the Blade-4 group and 365 in the Blade-3 group. The minimum detectable effect size, under the assumption of an 85% FPS rate for the Blade-3 group, was 5.3%, with a significance level of 0.05 and a power of 0.8. The FPS rates for the Blade-3 and Blade-4 groups in our study were 84.9% and 86.8%, respectively, prior to PSM, and 84.8% and 87.3%, respectively, after PSM. Therefore, these differences were not statistically significant.”

C) In the introduction part, line 63-64, there must be more factors affecting complications' risks of fail intubation, not only the number of attempt.

A) The manuscript has been revised to include several factors associated with endotracheal intubation-related complications.  

“ETI are associated with a high risk of adverse events including aspiration, cardiovascular collapse, severe hypoxemia, and cardiac arrest [8,9]. Several variables were associated with major peri-intubation adverse events. This involves not only patient-related factors, such as age, medical history, pre-intubation hemodynamics, and indications for ETI but also a multitude of procedure-related factors, including the operator's level of expertise, medications, and devices utilized during ETI [10,11]. In addition, several studies have shown the importance of achieving FPS, as the risk of adverse events significantly increases with the increasing number of ETI attempts [12,13]. Consequently, considerable efforts have been made to identify techniques and measures associated with improving the FPS rate.”

C) Moreover, the knowledge gap in this area was not clear, the DL has been introduced for a very long time why should this manuscript be published. Please elaborate more on why do we need to add this information into the scientific knowledge?

A) The rationale for establishing an association between the blade size of a video laryngoscope (VL) and the first-pass success (FPS) rate of endotracheal intubation (ETI) is elaborated in more detail in the revised manuscript. Several studies have shown that the selection of a specific blade size may be a simple method to increase the FPS rate. A knowledge gap exists as only Macintosh-type direct laryngoscopes (DL), not VLs, have been utilized in previous studies. In contrast to the DL, the VL reduces the need for axial alignment by utilizing an image sensor situated in the distal section of the blade, which offers a comprehensive view of the glottis. Consequently, extrapolating data related to the success rate of ETI and DL blade size to the VL may not be suitable. Recently, there has been a significant surge in the adoption of VLs as standard devices in clinical practice. Therefore, determining whether the blade size influences the efficacy of ETI via VL may be clinically relevant.

“Currently, there is a lack of data regarding the selection of blade size, forcing clinicians to rely on a combination of device availability, patient-specific characteristics, and the preferences of individual physicians. Recently, some studies have suggested that increasing the FPS rates through the selection of an appropriate blade size could be a straightforward approach that does not require additional interventions [14-16]. However, these studies solely assessed the Macintosh-type DL, and no study has examined the link between the success rate of ETI and VL blade size. A distinction exists in the way each equipment is operated between both approaches. DL requires displacement of the upper airway tissues to establish a straight line of sight extending from the oral cavity to the laryngeal inlet, while VL obviates the need to align the axes by utilizing an image sensor located in the distal portion of the blade to provide a panoramic view of the glottis. Therefore, it may be unsuitable to extrapolate data from the DL to the VL, representing a knowledge gap regarding the relationship between the FPS rate and blade size of the VL. Furthermore, it is of clinical significance to establish this association, considering the increasing prevalence of VL as a standard device for ETI.”

  1. C) Line 74, which types of blades did this manuscript focus on? VL or DL, some places you mentioned were DL, some were VL. Please revise to make the manuscript more congruent.
  2. A) The manuscript has been revised in response to the reviewers’ concerns. We focused on video laryngoscopes since their use has increased rapidly, and they are emerging as standard equipment in the clinical field. In addition, no study has evaluated the association between the blade size of the video laryngoscope and the first-pass success rate of endotracheal intubation.

“Currently, there is a lack of data regarding the selection of blade size, forcing clinicians to rely on a combination of device availability, patient-specific characteristics, and the preferences of individual physicians. Recently, some studies have suggested that increasing the FPS rates through the selection of an appropriate blade size could be a straightforward approach that does not require additional interventions [14-16]. However, these studies solely assessed the Macintosh-type DL, and no study has examined the link between the success rate of ETI and VL blade size. A distinction exists in the way each equipment is operated between both approaches. DL requires displacement of the upper airway tissues to establish a straight line of sight extending from the oral cavity to the laryngeal inlet, while VL obviates the need to align the axes by utilizing an image sensor located in the distal portion of the blade to provide a panoramic view of the glottis. Therefore, it may be unsuitable to extrapolate data from the DL to the VL, representing a knowledge gap regarding the relationship between the FPS rate and blade size of the VL. Furthermore, it is of clinical significance to establish this association, considering the increasing prevalence of VL as a standard device for ETI.”

Reviewer 2 Report

Comments and Suggestions for Authors

Single centre retrospective study assessing the effect of blade size on first pass success (FPS) rate of endotracheal intubation using the C-MAC video laryngoscope. The authors conclude that the FPS did not vary between blade 3 and 4 groups. Difficult glottis and complication rates were higher in the blade 3 group.

Limitations: Two major limitations, i) Retrospective study design, ii) non randomization, both mentioned in the limitations. 

Further limitation, which need to be discussed and mentioned in the eponymous section are relatively short patients, i.e. 162 cm tall, which means 10-15 cm shorter than adult people of several other countries. Also, the number of obese patients was very low compared to some other industrialized countries. Thus, the findings of this study have to be reproduced in countries with taller and a higher percentage of obese patients. 

Author Response

C) Further limitation, which need to be discussed and mentioned in the eponymous section are relatively short patients, i.e. 162 cm tall, which means 10-15 cm shorter than adult people of several other countries. Also, the number of obese patients was very low compared to some other industrialized countries. Thus, the findings of this study have to be reproduced in countries with taller and a higher percentage of obese patients. 

 (A) Thank you for your comments. To address your concerns, we revised the limitations section.

“Finally, the study population was relatively short in height and comprised a small proportion of obese patients[16]. Consequently, the applicability of our findings to other settings in which a substantial proportion of patients are taller or obese may be limited. To address these concerns, this study should be replicated in areas with a significant proportion of tall and obese patients.“

Reviewer 3 Report

Comments and Suggestions for Authors

The study lacked novelty; a propensity score match was suggested for statistical analysis; more scientific sound and novelty were needed.

Author Response

C) The study lacked novelty; a propensity score match was suggested for statistical analysis, and more scientific sounds and novelty are needed.

 (A) Thank you for your comments. Previous studies that evaluated the effect of blade size on the success rate of endotracheal intubation used only Macintosh-type direct laryngoscopes. To the best of our knowledge, no research has been conducted on the relationship between the first-pass success rate and video laryngoscope blade size. Recently, the use of video laryngoscopes as standard devices has significantly increased. Therefore, it may be clinically relevant to investigate whether the blade size influences the outcome of tracheal intubation using video laryngoscopy. Our study is unique in that it is the first to examine the relationship between the blade size of a video laryngoscope and the first-pass success rate based on available data. We have emphasized these points in the revised manuscript.

In addition to propensity score matching (PSM), various methodswere applied to reduce potential bias. Univariable and multivariable logistic regression analyses were performed to evaluate the relationship between the blade size and FPS rate. In addition, propensity score-adjusted models were built in three alternative ways: (1) regression adjustment by including the propensity score as a covariate in the regression model, (2) use of the propensity score to create stabilized weights, defined as the inverse probability of treatment weighting, and (3) propensity score matching (conditional logistic regression was performed to assess the association between blade size and FPS using 1:1 PSM data). Subgroup analyses were also conducted. We hope that our research may contribute to filling some of the knowledge gaps regarding the association between blade size and FPS rate in videolaryngoscope.

In introduction,

“Currently, there is a lack of data regarding the selection of blade size, forcing clinicians to rely on a combination of device availability, patient-specific characteristics, and the preferences of individual physicians. Recently, some studies have suggested that increasing the FPS rates through the selection of an appropriate blade size could be a straightforward approach that does not require additional interventions [14-16]. However, these studies solely assessed the Macintosh-type DL, and no study has examined the link between the success rate of ETI and VL blade size. A distinction exists in the way each equipment is operated between both approaches. DL requires displacement of the upper airway tissues to establish a straight line of sight extending from the oral cavity to the laryngeal inlet, while VL obviates the need to align the axes by utilizing an image sensor located in the distal portion of the blade to provide a panoramic view of the glottis. Therefore, it may be unsuitable to extrapolate data from the DL to the VL, representing a knowledge gap regarding the relationship between the FPS rate and blade size of the VL. Furthermore, it is of clinical significance to establish this association, considering the increasing prevalence of VL as a standard device for ETI.”

In Method,

“A logistic regression model was used to estimate the propensity score for each patient, representing the predicted probability of the blade size-4 group. We performed 1:1 matching using the nearest-neighbor greedy matching method with a caliper of 0.2.

Univariable and multivariable logistic regression analyses were performed to evaluate the relationship between blade size and FPS rate. The following variables were included as confounding factors: age, height, weight, BMI grade, sex, method of ETI, presence of anticipated difficult airway characteristics, and level of intubator.

Subsequently, the cohort was divided into five strata according to the quintiles of the estimated propensity scores. Stratified logistic regression analysis was performed to produce an overall OR considering the strata. Logistic regression analyses were performed separately within each stratum to examine the relationship between the blade size and FPS rate.

In addition, logistic regression models were built using propensity scores to adjust for differences among groups in three alternative ways: (1) regression adjustment by including the propensity score as a covariate in the regression model; (2) use of the propensity score to create stabilized weights, defined as the inverse probability of treatment weighting; and (3) propensity score matching (conditional logistic regression was performed to assess the association between blade size and FPS using 1:1 PSM data).

Subgroup analyses were also conducted. Subgroups were established based on the following criteria: age (<65 years old vs. ≥65 years old), sex (female vs. male), height (<170 cm vs. ≥170 cm; <175 cm vs. ≥175 cm), presence of difficult airway characteristics for laryngoscopy and ETI (yes vs. no), and level of intubator (junior vs. non-junior). For subgroup analysis, the intubators were divided into two categories: junior (first- or second-year residents) and non-junior (third- or fourth-year residents and EM specialists). During the subgroup analysis, the following variables were adjusted within each subgroup: age, height, weight, BMI grade, sex, methods for ETI, presence of anticipated difficult airway characteristics, and level of intubator. The variables corresponding to each subgroup were eliminated from the adjustments.”

In discussion,

“Based on the available data, this is the first study on the relationship between the blade size of a VL and the FPS rate.”

Round 2

Reviewer 1 Report

Comments and Suggestions for Authors

Dear Authors,

Thank you for thoroughly revising the manuscript.

I think it was very much improved and proper for publication.

Comments on the Quality of English Language

Very good English written language

Reviewer 3 Report

Comments and Suggestions for Authors

Authors did a great improvment: very good